# The Impact of Perioperative Fluid Balance on Postoperative Complications after Esophagectomy for Esophageal Cancer

**DOI:** 10.3390/jcm11113219

**Published:** 2022-06-05

**Authors:** Yuto Kubo, Koji Tanaka, Makoto Yamasaki, Kotaro Yamashita, Tomoki Makino, Takuro Saito, Kazuyoshi Yamamoto, Tsuyoshi Takahashi, Yukinori Kurokawa, Masaaki Motoori, Yutaka Kimura, Kiyokazu Nakajima, Hidetoshi Eguchi, Yuichiro Doki

**Affiliations:** 1Department of Gastroenterological Surgery, Graduate School of Medicine, Osaka University, Osaka 565-0871, Japan; ykubo@grsurg.med.osaka-u.ac.jp (Y.K.); yamasakm@hirakata.kmu.ac.jp (M.Y.); kyamashita@gesurg.med.osaka-u.ac.jp (K.Y.); tmakino@gesurg.med.osaka-u.ac.jp (T.M.); tsaito@gesurg.med.osaka-u.ac.jp (T.S.); kyamamoto13@gesurg.med.osaka-u.ac.jp (K.Y.); ttakahashi2@gesurg.med.osaka-u.ac.jp (T.T.); ykurokawa@gesurg.med.osaka-u.ac.jp (Y.K.); knakajima@gesurg.med.osaka-u.ac.jp (K.N.); heguchi@gesurg.med.osaka-u.ac.jp (H.E.); ydoki@gesurg.med.osaka-u.ac.jp (Y.D.); 2Department of Surgery, Osaka General Medical Center, Osaka 558-8558, Japan; mmotoori@gh.opho.jp; 3Department of Surgery, Kindai University Nara Hospital, Nara 630-0293, Japan; you-kimura@med.kindai.ac.jp

**Keywords:** fluid balance, fluid overload, postoperative complication, esophageal cancer, surgery, esophagectomy

## Abstract

Background: Perioperative fluid balance is an important indicator in the management of esophageal cancer patients who undergo esophagectomy. However, the association between perioperative fluid balance and postoperative complications after minimally invasive esophagectomy (MIE) remains unclear. Methods: This study included 115 patients with thoracic esophageal squamous cell cancer who underwent MIE between January 2018 and January 2020. We retrospectively evaluated the association between perioperative fluid balance from during surgery to postoperative day (POD) 2, and postoperative complications. Results: The patients were divided into lower group and higher group based on the median fluid balance during surgery and at POD 1 and POD 2. We found that the higher group at POD 1 (≥3000 mL) was the most important indicator of postoperative complications, such as acute pneumonia within 7 days after surgery, and anastomotic leakage (*p* = 0.029, *p* = 0.024, respectively). Moreover, the higher group at POD 1 was a significant independent factor for acute postoperative pneumonia by multivariate analysis (OR: 3.270, 95% CI: 1.077–9.929, *p* = 0.037). Conclusion: This study showed that fluid overload at POD 1 had a negative influence on postoperative complications in patients with esophageal cancer. The fluid balance must be strictly controlled during the early postoperative management of patients undergoing esophageal cancer surgery.

## 1. Introduction

Esophageal cancer is the eighth most common cancer worldwide, and has the sixth leading worst prognosis. The number of patients with esophageal cancer has been increasing over the past 30 years [1].

According to an analysis by the Esophageal Complications Consensus Group (ECCG), the overall incidence of postoperative complications after esophagectomy was 65%. In particular, the most common complication after esophagectomy in esophageal cancer was pneumonia (29%), and the second most common complication was leakage from anastomosis (19%) [2]. The rate of overall complications after gastrectomy was 42%, and the most common complications were pneumonia (12%) and anastomotic leakage (9%) [2]. Also, laparoscopic colectomy for colon cancer had 21% overall postoperative complications, including wound infection (4%), anastomotic failure (3%) and pneumonia (2%) [3]. Therefore, esophagectomy for esophageal cancer has a higher incidence of postoperative complications than other gastrointestinal cancer resections.

It has been recognized that postoperative complications in patients with various types of cancer negatively impact prognoses [4,5]. Moreover, a previous study in the Japan Clinical Oncology Group (JCOG) showed that postoperative infectious complications after esophagectomy may impact prognosis and be an independent risk factor for decreased overall survival in patients with esophageal cancer [6]. Therefore, treatment strategies, such as surgical procedures and perioperative management, should be carefully considered in order to prevent postoperative pneumonia and anastomotic leakage.

Perioperative fluid balance combined vasopressor is an important indicator in the postoperative management of patients after esophageal cancer surgery. Several studies showed that intraoperative and postoperative fluid overload was a risk factor for adverse surgical outcomes, including mortality, pulmonary morbidity, anastomotic leakage, and cardiac complications in patients who underwent esophageal cancer surgery;most patients included in these studies underwent right-sided thoracotomy [7,8,9].

The number of minimally invasive esophagectomy (MIE) methods, such as transhiatal esophagectomy and thoracoscopic esophagectomy, is increasing with the development of techniques and devices. Previous studies have shown that MIE may be a less invasive surgical procedure than open thoracotomy, and may be superior to open thoracotomy as a result of fewer postoperative complications and lower in-hospital mortality [10,11,12]. However, the association between perioperative fluid balance in MIE and postoperative complications remains unclear, especially with the development of MIE in recent years.

Therefore, the aim of the present study was to investigate whether perioperative fluid balance influences postoperative complications in patients with esophageal cancer who underwent surgery in recent years with the increasing use of MIE.

## 2. Materials and Methods

### 2.1. Patients

Of the patients with thoracic esophageal cancer who underwent subtotal esophagectomy at Osaka University between January 2018 and January 2020, 132 patients were included in the present study. Of these 132 patients, there were 2 patients who underwent cervical esophageal resection, 3 patients who underwent surgery for recurrence after radical chemoradiotherapy, and 12 patients who underwent right-sided thoracotomy that were excluded; therefore, 115 patients were included in the analysis. Among them, 26 patients underwent upfront esophagectomy, 81 underwent neoadjuvant chemotherapy followed by surgery, and 8 underwent neoadjuvant chemoradiotherapy. The stage of patients was judged using computed tomography (CT) and endoscopy, before and after neoadjuvant treatment, along with 18F-fluorodeoxyglucose-positron emission tomography (^18^F-FDG-PET) scans, when available. Clinicopathological findings were classified based on the Union for International Cancer Control TNM classification (8th edition) [13]. The clinical responses were determined according to the criteria of the World Health Organization response criteria using CT, endoscopy, and ^18^F-FDG-PET [14]. Additionally, the comorbidities of the patients were quantified by the Charlson Comorbidity Index (CCI), which is a commonly used score that quantifies multiple comorbidities [15].

### 2.2. Fluid Balance

Perioperative fluid balance in this study was calculated as fluid administered and eliminated through all routes, including blood loss and drainage, and was recorded from during surgery to postoperative day (POD) 2 during the ICU stay. The total fluid balance at POD 1 was calculated from during surgery to POD 1, and the total fluid balance at POD 2 was calculated from during surgery to POD 2. The fluid balance was calculated by subtracting the fluid eliminated from the total fluid administered.

### 2.3. Neoadjuvant Therapy and Surgical Procedure

At our hospital, neoadjuvant chemotherapy followed by surgery was performed for patients with cStage I with lymph metastasis (T1N1M0), II, III, or IV esophageal cancer without distant organ metastasis. Additionally, patients whose tumors could invade adjacent organs, including the trachea or aorta, underwent neoadjuvant chemoradiotherapy followed by surgery. The neoadjuvant chemotherapy regimen consisted of 5-FU and cisplatin plus docetaxel or 5-FU, and cisplatin plus doxorubicin or 5-FU and cisplatin [16]. Two or three courses of chemotherapy were administered with a 2–3 week rest period. The neoadjuvant chemoradiotherapy regimen consisted of 5-FU, cisplatin and simultaneous 40–60 Gy radiation. The patients with esophageal cancer underwent surgery 3–8 weeks after neoadjuvant therapy was completed.

The surgical approach used for thoracic esophageal cancer was subtotal esophagectomy. The selection of thoracoscopic surgery or robotic surgery was made according to the patient’s general condition, considering age, comorbidities, and TNM classification. In lymph node dissection, patients with lower or middle esophageal cancer underwent two-field lymph node dissection, and patients with upper esophageal cancer or metastases to cervical or recurrent nerve lymph nodes underwent three-field lymph node dissection, regardless of the esophageal cancer location [17]. All patients had chest drains inserted, and the patients with three-field lymph node dissection had cervical drains placed.

### 2.4. Perioperative Management

The patients in this study received only general anesthesia, or a combination of general anesthesia and epidural analgesia, unless otherwise contraindicated. The postoperative protocol for epidural analgesia was levobupivacaine 0.167% at continuous infusion of 4 mL/h for 5 days after surgery. All patients were injected with a peripherally inserted central venous catheter (PICC). These patients were kept in a surgical intensive care unit (ICU) for at least 2 days after esophageal cancer surgery, and removed from ventilators at POD1. The choices of fluid management, anesthetic agent, and vasopressor or inotropic agent administration were determined by the anesthesiologist and ICU doctor, who monitored vital signs such as electrocardiogram, continuous blood pressure using an arterial line, pulse oximetry (SpO2), urine output, and respiratory ventilation. Arterial blood gas analyses were performed intermittently. A feeding tube (gastrostomy or jejunostomy tube) was basically placed during surgery. Enteral nutrition was started at 10 mL/h on postoperative day 1 (POD1) and was gradually increased every day thereafter. Patients stopped the intake of fluids until POD 4, and initiated fluids after POD 5, with solids after POD 6–7.

### 2.5. Postoperative Complications

The complications during course from postoperative to discharge were defined as Clavien–Dindo classification grade ≥ II [18]. Pneumonia was defined as new pulmonary infiltrates with clinical evidence of an infectious origin, such as new fever, purulent sputum, leukocytosis, and decreased oxygenation [19]. In addition, postoperative pneumonia was divided on the basis of the date of onset: pneumonia in the acute phase, which occurred before or at POD 7 (acute pneumonia), and pneumonia in the subacute phase, which occurred 8 days after surgery or later (subacute pneumonia), as previously described [20,21]. Arrythmia in our study included atrial fibrillation (AF), which is one of the most common arrythmias in esophagectomy, and AF was defined as an absent P wave before the QRS complex, with an irregular ventricular rhythm on the rhythm strips [22]. Anastomotic leakage was judged based on CT, endoscopy, and esophagography. Recurrent nerve palsy was diagnosed by bronchoscopy or symptoms of hoarseness [20], and sputum excretion difficulty was determined by cricothyroidotomy and suction for sputum [23]. The diagnosis of chylothorax was made based on the clinical quantity or quality of chest drain output, by either a change in the quality of chest drainage to a milky appearance regardless of chest tube output, or by pleural fluid triglycerides >110 mg/dL [24]. Pulmonary embolism was determined by decreased oxygenation, D-dimer measurement, and computed tomography pulmonary angiography [25]. Surgical site infection (SSI) was defined as spontaneous or surgically opened purulent discharge with positive cultures. Moreover, acute kidney injury (AKI) was defined by increased serum creatinine (Cr) and urine output using Kidney Disease Improving Global Outcomes (KDIGO) clinical practice guidelines [26].

### 2.6. Statistical Analysis

The Mann–Whitney U test, χ^2^ test, or Student’s *t*-test was used to compare patient characteristics, preoperative treatment details, intraoperative factors, and postoperative course. Cox proportional hazards regression models were used to identify the variables significantly associated with postoperative complications. The corresponding HRs were calculated, along with the 95% CIs. Continuous variables are expressed as the mean ± SD, unless otherwise stated. Statistical significance was indicated by *p* value < 0.05. All analyses were performed using JMP^®^ 14 (SAS Institute Inc., Cary, NC, USA).

## 3. Results

### 3.1. Perioperative Fluid Management

Figure 1 shows the perioperative fluid balance from during surgery to POD 2. The median fluid balance during surgery, at POD 1, and POD 2, were 2585 mL, 3036 mL, and 2744 mL, respectively. A subsequent analysis was performed by dividing the patients into the two groups during surgery and at POD 1 and POD 2, using the median fluid balance at each time (during surgery; 2600 mL, POD 1; 3000 mL, POD 2; 2700 mL): a group with a fluid balance of less than the median (lower group), and a group with a fluid balance of the median or more (higher group). Among the 115 patients included in this study during surgery and at POD 1 and POD 2, the lower group included 58, 56, and 55 patients, and the higher group included 57, 59, and 60 patients, respectively.

### 3.2. Influence of Perioperative Fluid Management on Postoperative Complications

There was no difference in overall complications during course from postoperative to discharge (Clavien–Dindo classification grade ≥ II) between the two groups during surgery, at POD 1, and POD 2, respectively. The overall pneumonia rate was not different between the lower group and the higher group. However, acute pneumonia was significantly higher in the higher group than in the lower group at only POD 1 (12.5% vs. 28.8%, *p* = 0.029), although subacute pneumonia showed no difference between the two groups. Moreover, anastomotic leakage and sputum excretion difficulty was significantly higher in the higher group than in the lower group at only POD 1 (1.8% vs. 11.9%; *p* = 0.024, 1.8% vs. 10.2%; *p* = 0.048, respectively). There was no difference in arrythmia, recurrent nerve palsy, chylothorax, pulmonary embolism, or AKI (KDIGO clinical practice guidelines Stage ≥ 1) between the two groups. Furthermore, no patients with acute kidney failure underwent dialysis after surgery. The higher perioperative group had a higher SSI rate than the lower perioperative group (Table 1). Also, the cutoff values using ROC curves for the fluid balance at POD1 in postoperative acute pneumonia and anastomotic leakage were 3080 mL and 3094 mL, respectively, which were similar to median values (acute pneumonia: sensitivity of 0.71 and specificity of 0.44; anastomotic leakage: sensitivity of 0.88 and specificity of 0.46) (Appendix A).

### 3.3. Patient Characteristics

The following analyses were evaluated using only fluid balance at POD 1, since the higher group at POD 1 was an important predictor of postoperative complications such as acute pneumonia, anastomotic leakage, sputum excretion difficulty, and SSI, according to the results of this study. Table 2 summarizes the characteristics of patients in the lower group and those in the higher group, based on POD 1 fluid balance. There were no significant differences in age, body mass index (BMI), body surface area (BSA), American Society of Anesthesiology physical status (ASA-PS), esophageal cancer location, cStage, and histological type between the two groups. In addition, the preoperative Alb level was not different between the lower group and the higher group, and the CCI was similar in the two groups. There was no significant difference in preoperative treatment, such as neoadjuvant chemotherapy or neoadjuvant chemoradiotherapy, between the two groups. However, the higher group had a significantly higher proportion of males than the lower group.

### 3.4. Fluid Balance and Surgical Factors

The association between fluid balance and intraoperative factors between the two groups is shown in Table 3. Neither the surgical method (thoracoscopy or robotic surgery) nor reconstruction organ had a significant influence on POD 1 fluid balance, and there was no variation in the reconstruction route and feeding tube in the two groups. On the other hand, the higher group at POD 1 had more patients with three fields of lymph node dissection than the lower group. Additionally, the operative time was significantly higher in the higher POD 1 group than in the lower POD 1 group. Bleeding volume during surgery was not different between the two groups, and there was no significant difference between the higher group and lower group in the rate of thoracic duct resection.

### 3.5. Fluid Balance and Postoperative Course

Table 4 shows the relationship between fluid balance and the postoperative course at POD 1 during the ICU stay. The total infusion volume at POD 1, including crystalloid, colloidal solution, and blood transfusions, was significantly higher in the higher group than in the lower group, although the tube feeding volume via gastrostomy or jejunostomy tube was similar between the two groups. Moreover, the higher group had significantly greater weight increase rate at POD1 than in the lower group. On the other hand, the higher group had significantly less urine output than the lower group, although there was no difference in drainage output between the two groups. There was no difference in blood transfusion, vasopressor, and diuretics utilization rates between the two groups. Furthermore, the vasopressor utilization did not impact on infusion volume and total fluid balance at POD1 (*p* = 0.860, *p* = 0.189). The higher group had no difference in CRP at POD 1 compared with the lower group.

### 3.6. The Association between Perioperative Fluid Management and Postoperative Complications

Table 5 indicates the risk factors associated with postoperative acute pneumonia by univariate and multivariate analyses. For the higher group at POD 1, CCI ≥ 2, postoperative sputum excretion difficulty and recurrent nerve palsy (Clavien–Dindo Grade ≥ II) were significant independent factors for acute postoperative pneumonia by multivariate analysis (higher group; OR: 3.270, 95% CI: 1.077–9.929, *p* = 0.037, CCI; OR: 4.191, 95% CI: 1.222–14.37, *p* = 0.027, postoperative sputum excretion difficulty; OR: 6.337, 95% CI: 1.160–34.60, *p* = 0.033, postoperative recurrent nerve palsy; OR: 5.900, 95% CI: 1.571–22.16, *p* = 0.009). Furthermore, the higher group at POD 1 and low Alb (<3.0 g/dL) tended to have risk factor for anastomotic leakage after surgery, according to multivariate analyses (Table 6).

## 4. Discussion

This study found that fluid overload had a negative association with postoperative complications, such as pneumonia and anastomotic leakage, in patients with esophageal cancer who underwent surgery. In particular, an overload of fluid balance at POD 1 in the perioperative period had the most negative impact on both postoperative pneumonia and anastomotic leakage after esophagectomy for esophageal cancer.

Yibulayin W et al. reported that intraoperative blood loss, postoperative overall complications, and in-hospital mortality were significantly lower in MIE than in open esophagectomy, in a meta-analysis study [12]. Additionally, a previous multicenter, open-label, randomized controlled trial, including 115 patients with resectable cancer of the esophagus or gastroesophageal junction, showed that the patients who underwent MIE had fewer postoperative pulmonary infections, such as pneumonia and bronchopneumonia, within 2 weeks after surgery than those who underwent open esophagectomy (relative risk; 0.30, 95% CI; 0.12–0.76, *p* = 0.005) [11]. In this study, the overall postoperative complication rate was lower in MIE than in open esophagectomy (57.0% vs. 81.8%, *p* = 0.096). Additionally, patients undergoing MIE had significantly lower postoperative serum CRP levels than patients undergoing open esophagectomy, according to a propensity score-matched analysis or prospective study [27,28]. Therefore, we suggest that MIE may be a less invasive surgical procedure than open thoracotomy.

A previous study showed that goal-directed fluid therapy, which targets continuously measured hemodynamic variables, such as cardiac output, stroke volume, and pulse pressure variation, decreases fluid balance compared with normal infusion therapy; such a therapy also reduces inflammatory reactions, including TNF-α and IL-6, after lung cancer surgery [29]. Recently, it was shown that perioperative restrictive fluid management, which is defined as a near-zero perioperative fluid balance or referred to as a zero-balance approach, may also be superior or equivalent to goal-directed fluid therapy [30]. Moreover, it may reduce additional costs and resource utilization. Several studies have shown that restrictive fluid management was superior to standard fluid management, since it prevented postoperative complications in clinical trials and meta-analyses of abdominal surgery [31,32,33]. Restrictive fluid management has been widely recommended and incorporated in enhanced recovery after surgery (ERAS) programs, and constitutes an important element of these programs [34]. Therefore, it is important to restrict the perioperative fluid balance for patients with esophageal cancer undergoing esophagectomy.

To the best of our knowledge, only one study has revealed the relationship between perioperative fluid balance and postoperative complications in patients who underwent MIE for esophageal cancer in recent years. In our study, most of the included patients underwent thoracoscopic esophagectomy, and the results suggested that an overload of fluid balance (≥3000 mL) at POD 1, but not on the day of operation, may be a significant independent factor for postoperative pneumonia after esophageal cancer surgery. Moreover, a higher fluid balance (≥3000 mL) at POD 1 had a significantly negative influence on anastomotic leakage. Additionally, it was possible that the fluid balance at POD 1 in this study (3000 mL overload) was lower than the balance reported by previous studies that included patients who underwent right-sided thoracotomy (6900–7873 mL overload) [8,9]. Based on these results, we suggest that fluid balance in patients with esophageal cancer needs to be strictly controlled after MIE, especially at POD 1.

The reason that perioperative fluid balance overload is associated with worse complications in patients with esophageal cancer could be that excessive fluid administration may increase extravascular fluid in the tissue of the lung, which can induce pulmonary edema. Therefore, oxygen exchange may be inhibited, which increases the risk of postoperative respiratory failure and pneumonia [35]. Moreover, increased water in the body due to fluid overload can result in edema around the anastomosis, and induce leakage [36]. Indeed, in this study, the higher fluid balance group had significantly higher rates of both postoperative pneumonia and anastomotic leakage than the lower fluid balance group.

Several studies have shown that postoperative complications negatively affect the prognosis of patients undergoing esophageal cancer surgery. Tanaka et al. reported that the disease-free survival (DFS) of patients with acute pneumonia within 7 days after esophagectomy was significantly poorer than that of other patients, although pneumonia occurring after 8 days postoperatively did not influence DFS [20]. Furthermore, anastomotic leakage after esophageal cancer surgery was a significant independent prognostic factor for OS and recurrence-free survival (RFS), according to multivariate analysis [37]. A previous study reported that systemic inflammation contributed to the proliferation and invasion of cancer cells [38,39], and it may induce residual cancer cell growth [40]. Hence, treatment strategies, such as surgical procedures and perioperative management, must be carefully considered in order to prevent postoperative pneumonia, anastomotic leakage, and causes of systemic inflammation, which may have a negative impact on the survival of patients with esophageal cancer who undergo surgery.

Regarding surgical factors, this study showed that the fluid balance at POD 1 was associated with operative time. Hence, we suggest that fluid balance must be strictly managed for patients who have undergone esophagectomy with long operative times. Additionally, in postoperative management, urine output was significantly lower in the higher fluid group at POD 1, although diuretic use was similar between the two groups. Based on these results, it is suggested that perioperative fluid management should consider not administering excessive fluids when urine output is low.

There are several limitations to our study, which investigated the associations between the perioperative fluid balance and postoperative complications. For example, this study was a retrospective study in a single institution. In order to confirm the results of this study, large-scale prospective studies need to be conducted at multiple centers. Secondly, anesthesiologists and ICU doctors who participated in this study were random, not constant. Hence, indications for fluid management or blood products differed by anesthesiologists and ICU doctors. The fluid balance may differ by the anesthesiologists. Thirdly, this study did not evaluate a total volume of epidurals, including bolus during pain, in perioperative fluid balance.

## 5. Conclusions

The present study showed that fluid overload at POD 1 had a negative influence on postoperative pneumonia and anastomotic leakage. This result suggests that we need to strictly manage fluid balance in the early postoperative management of patients with esophageal cancer undergoing surgery.

## Figures and Tables

**Figure 1 jcm-11-03219-f001:**
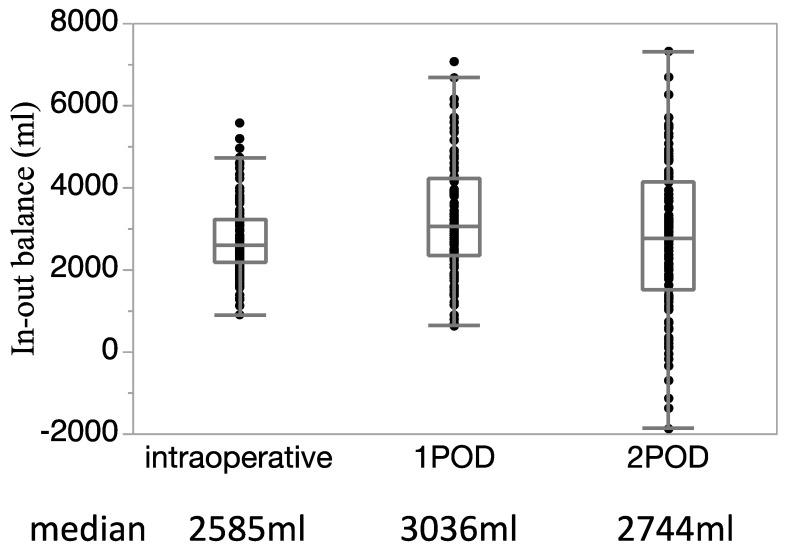
Perioperative fluid balance from during surgery to postoperative day 2. The median fluid balance during surgery, at POD 1, and POD 2, were 2585 mL, 3036 mL, and 2744, respectively.

**Table 1 jcm-11-03219-t001:** Postoperative complications by fluid balance in the patients.

	Intraoperative Fluid Balance		*p* Value	POD 1 Fluid Balance		*p* Value	POD 2 Fluid Balance		*p* Value
	Lower	Higher		Lower	Higher		Lower	Higher	
No. of patients	58 (50.4%)	57 (49.6%)		56 (48.7%)	59 (51.3%)		55 (47.8%)	60 (52.2%)	
Postoperative complication *	34 (58.6%)	28 (49.1%)	0.307	31 (55.4%)	31 (52.5%)	0.762	32 (48.1%)	30 (50.0%)	0.379
pneumonia	17 (29.3%)	18 (31.6%)	0.792	15 (26.8%)	20 (33.9%)	0.407	16 (29.1%)	19 (31.7%)	0.764
acute pneumonia ^(a)^	9 (15.5%)	15 (26.3%)	0.153	7 (12.5%)	17 (28.8%)	0.029	8 (14.6%)	16 (26.7%)	0.107
subacute pneumonia ^(b)^	8 (13.8%)	3 (5.3%)	0.114	8 (14.3%)	3 (5.1%)	0.089	8 (14.6%)	3 (5.0%)	0.078
arrhythmia	7 (12.1%)	5 (8.8%)	0.562	6 (10.7%)	6 (10.1%)	0.924	7 (12.7%)	5 (8.3%)	0.441
anastomotic leakage	4 (6.9%)	4 (7.0%)	0.980	1 (1.8%)	7 (11.9%)	0.024	2 (3.6%)	6 (10.0%)	0.170
recurrent nerve palsy	6 (10.3%)	6 (10.5%)	0.975	6 (10.7%)	6 (10.1%)	0.924	5 (9.1%)	7 (11.7%)	0.651
sputum excretion difficulty	2 (3.5%)	5 (8.8%)	0.226	1 (1.8%)	6 (10.2%)	0.048	2 (3.7%)	5 (8.3%)	0.284
chylothorax	2 (3.5%)	3 (5.3%)	0.632	3 (5.4%)	2 (3.4%)	0.604	3 (5.5%)	2 (3.3%)	0.577
pulmonary embolism	1 (1.7%)	1 (1.8%)	0.990	0	2 (3.4%)	0.100	2 (3.6%)	0	0.084
SSI	0	4 (7.0%)	0.016	0	4 (6.8%)	0.019	0	4 (6.8%)	0.021
other	4 (6.9%)	3 (5.3%)	0.714	4 (7.1%)	3 (5.1%)	0.644	4 (7.3%)	3 (5.0%)	0.611
Acute kidney injury **	1 (1.7%)	1 (1.8%)	0.990	0	2 (3.4%)	0.100	0	2 (3.3%)	0.105

POD: postoperative day, * Clavien–Dindo classification grade ≥ II, ** Kidney Disease Improving Global Outcomes clinical practice guidelines Stage ≥ 1, SSI: surgical site infection, ^(a)^ within 7 days after surgery, ^(b)^ 8 days after surgery.

**Table 2 jcm-11-03219-t002:** Characteristics of the patients in POD1 fluid balance.

		POD 1 Fluid Balance		*p* Value
	All Patients	Lower	Higher	
No. of patients	115	56 (48.7%)	59 (51.3%)	
Age	68.7 ± 9.7	67.7 ± 9.1	69.7 ± 10.2	0.235
Sex				
male	79 (68.7%)	32 (57.1%)	47 (79.7%)	0.009
female	36 (31.3%)	24 (42.9%)	12 (20.3%)	
BMI	20.7 ± 3.2	20.3 ± 2.8	21.0 ± 3.4	0.354
BSA (m^2^)	1.58 ± 0.17	1.55 ± 0.16	1.61 ± 0.18	0.121
ASA-PS *				
class1	81 (70.4%)	39 (69.6%)	42 (71.2%)	0.740
class2	24 (20.9%)	13 (23.2%)	11 (18.6%)	
class3	10 (8.7%)	4 (7.1%)	6 (10.2%)	
Tumor location				
upper third	14 (13.5%)	3 (5.4%)	11 (18.6%)	0.072
middle third	48 (45.2%)	24 (42.9%)	24 (40.7%)	
lower third	53 (41.4%)	29 (51.8%)	24 (40.7%)	
cStage (TNM)				
I	18 (15.7%)	9 (16.1%)	9 (15.3%)	0.703
II	34 (29.6%)	15 (26.8%)	19 (32.2%)	
III	50 (43.5%)	27 (48.2%)	23 (39.0%)	
IV	13 (11.3%)	5 (8.9%)	8 (13.6%)	
Histological type				
squamous cell carcinoma	107 (93.0%)	51 (91.1%)	56 (94.9%)	0.416
adenocarcinoma	8 (7.0%)	5 (8.9%)	3 (5.1%)	
CCI				
score ≤ 1	98 (85.2%)	46 (82.1%)	52 (88.1%)	0.365
score ≥ 2	17 (14.8%)	10 (17.9%)	7 (11.9%)	
Alb (g/dL) *	3.5 ± 0.5	3.6 ± 0.5	3.5 ± 0.5	0.908
Preoperative treatment				
none	26 (22.6%)	10 (17.9%)	16 (27.1%)	0.233
chemotherapy and radiation chemotherapy	89 (77.4%)	46 (82.1%)	43 (72.9%)	

Data are expressed as mean ± SD. POD: postoperative day, * preoperative data, BMI: body mass index, BSA: body surface area, ASA-PS: American Society of Anesthesiology physical status, cStage: clinical stage, CCI: Charlson comorbidity index, Alb: albumin.

**Table 3 jcm-11-03219-t003:** Surgical factor and POD 1 fluid balance.

		POD 1 Fluid Balance		*p* Value
	All Patients	Lower	Higher	
No. of patients	115	56 (48.7%)	59 (51.3%)	
Surgery				
thoracoscopy	93 (80.9%)	48 (85.7%)	45 (76.3%)	0.196
robotic surgery	22 (19.1%)	8 (14.3%)	14 (23.7%)	
Field of lymph node dissection				
2 fields	60 (52.2%)	34 (60.7%)	26 (44.1%)	0.073
3 fields	55 (47.8%)	22 (39.3%)	33 (55.9%)	
Reconstruction route				
ante-thoracic	11 (9.6%)	3 (5.4%)	8 (13.6%)	0.172
retrosternal	68 (59.1%)	32 (57.1%)	36 (61.0%)	
posterior mediastinal	36 (31.3%)	21 (37.5%)	15 (25.4%)	
Reconstruction organ				
stomach tube	113 (98.3%)	56 (100%)	57 (96.6%)	0.100
other	2 (1.7%)	0	2 (3.4%)	
Feeding tube				
gastrostomy or jejunostomy	113 (98.3%)	56 (100%)	57 (96.6%)	0.100
none	2 (1.7%)	0	2 (3.4%)	
Operative Time (min)	479 ± 87	458 ± 85	499 ± 88	0.009
Bleeding Volume (mL)	155 ± 201	151 ± 249	160 ± 143	0.236
Thoracic duct				
resection	21 (18.3%)	11 (19.6%)	10 (17.0%)	0.709
preservation	94 (81.7%)	45 (80.4%)	49 (83.0%)	

Data are expressed as mean ± SD. POD: postoperative day.

**Table 4 jcm-11-03219-t004:** Postoperative course at POD 1 in the patients.

		POD 1 Fluid Balance		*p* Value
	All Patients	Lower	Higher	
No. of patients	115	56 (48.7%)	59 (51.3%)	
Infusion volume * (mL)	2937 ± 804	2663 ± 578	3198 ± 901	<0.001
Tube feeding volume (mL)	124 ± 56	118 ± 63	130 ± 47	0.140
Drainage output (mL)	839 ± 486	907 ± 632	774 ± 276	0.493
Urine output (mL)	1697 ± 745	1948 ± 803	1459 ± 601	0.001
Use of colloidal solution use				
Albumin	32 (27.7%)	17 (30.4%)	15 (25.4%)	0.555
Hydroxyethyl starch	85 (72.2%)	39 (69.6%)	44 (74.6%)	
Use of blood transfusion	18 (15.7%)	6 (10.7%)	12 (20.3%)	0.152
Use of vasopressor	62 (53.9%)	32 (57.1%)	30 (50.9%)	0.498
Use of diuretic	12 (10.5%)	4 (7.1%)	8 (13.8%)	0.243
Weight increase rate (%) **	3.4 ± 2.3	3.0 ± 2.0	3.8 ± 2.5	0.044
CRP (mg/L)	5.2 ± 2.6	4.9 ± 2.3	5.5 ± 2.8	0.198

Data are are expressed as mean ± SD, * including crystalloid, colloidal solution, and blood transfusion. ** weight change compared with preoperative weight, CRP: C-reactive protein.

**Table 5 jcm-11-03219-t005:** Factors associated with acute pneumonia after surgery by univariate and multivariate analyses.

Variables	Univariate Analysis			Multivariate Analysis		
	OR	95% CI	*p* Value	OR	95% CI	*p* Value
Age ≥ 65 (vs. <65)	2.222	0.695–7.104	0.178			
Sex male (vs. female)	1.963	0.669–5.760	0.219			
ASA-PS class2, 3 (vs. class 1)	1.584	0.615–4.079	0.341			
CCI ≥ 2 (vs. ≤1)	3.335	1.112–10.00	0.032	4.191	1.222–14.37	0.027
Thoracoscopic surgery (vs. Robotic surgery)	1.233	0.375–4.057	0.731			
Operative time ≥ 475 min (vs. <475 min)	1.068	0.434–2.262	0.886			
Field of lymph node dissection 3 fields (vs. 2 fields)	1.703	0.677–4.288	0.258			
POD 1 fluid balance higher group (vs. lower group)	2.833	1.072–7.488	0.036	3.270	1.077–9.929	0.037
Weight loss rate at POD1 ≥ 3.5% (vs. <3.5%)	1.465	0.588–3.650	0.412			
Postoperative sputum excretion difficulty CD classification Grade ≥ II (vs. ≤I)	5.867	1.216–28.30	0.028	6.337	1.160–34.60	0.033
Postoperative recurrent nerve palsy CD classification Grade ≥ II (vs. ≤I)	4.772	1.366–16.33	0.014	5.900	1.571–22.16	0.009

OR: odds ratio, CI: confidence interval, ASA-PS: American Society of Anesthesiology physical status, CCI: Charlson comorbidity index, POD: postoperative day, CD: Clavien–Dindo.

**Table 6 jcm-11-03219-t006:** Factors associated with anastomotic leakage after surgery by univariate and multivariate analyses.

Variables	Univariate Analysis			Multivariate Analysis		
	OR	95% CI	*p* Value	OR	95% CI	*p* Value
Age ≥ 65 (vs. <65)	2.855	0.337–24.18	0.336			
Sex male (vs. female)	3.403	0.402–28.74	0.261			
ASA-PS class2, 3 (vs. class 1)	1.471	0.331–6.533	0.612			
CCI ≥ 2 (vs. ≤1)	1.231	0.142–10.69	0.851			
Alb ≥ 3.0 g/dL (vs. <3.0 g/dL)	4.750	1.006–22.43	0.049	5.065	0.996–25.75	0.051
Operative time ≥ 475 min (vs. <475 min)	1.636	0.372–7.190	0.515			
Field of lymph node dissection 3 fields (vs. 2 fields)	1.703	0.677–4.288	0.258			
Reconstruction route posterior mediastinal (vs. other)	3.402	0.403–28.75	0.261			
POD 1 fluid balance higher group (vs. lower group)	7.404	0.880–62.26	0.065	7.739	0.895–68.89	0.063
Weight loss rate at POD1 ≥ 3.5% (vs. <3.5%)	1.667	0.379–7.337	0.499			

OR: odds ratio, CI: confidence interval, ASA-PS: American Society of Anesthesiology physical status, CCI: Charlson comorbidity index, POD: postoperative day.

## Data Availability

The data presented in this study are available on request from the corresponding author.

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
