# Peer review of "The Impact of Perioperative Fluid Balance on Postoperative Complications after Esophagectomy for Esophageal Cancer"

_jcm, 2022, doi:10.3390/jcm11113219_

Round 1
Reviewer 1 Report
Many thanks for the opportunity to review this manuscript and I commend the authors on taking on this topic. There are a few areas that I think need to be addressed in this manuscript.
Introduction: the introduction is well researched. However, there is minimal reference made to use of vasopressor agents. It is a balancing act between vasopressor use and fluid resuscitation in esophagectomy and this needs to be acknowledged.
Methods: My main issue here is around how perioperative management was recorded. Although use of epidurals is mentioned, there is no recording of this in the results. Also, there is no mention of intra-operative use of vasopressors (with duration and quantity) or post-operative use of vasopressors. Patients with epidurals have altered sympathetic tone and often end up fluid overloaded: one of the potential benefits of MIE is avoiding this situation.
Results: Cutting the groups in two based on the median seems a little arbitrary. It would be nice to consider a cutpoint analysis to try determine if there is a threshold where fluid overload is significant. Again, the lack of detail on vasopressors needs to be addressed. How many patients were on pressors intra-op and post op, for how long and what total quantity? How may had epidurals? The language needs clarification that it is the group who are overloaded on POD1 that have complications during their course (not that they have a complication on POD1 as the manuscript seems to suggest).
The discussion is fine, but will need to be rewritten taking into account the points above.
Author Response
Thank you for your e-mail of May 18, 2022. We are pleased to have a chance to resubmit our revised manuscript in Journal of Clinical Medicine.
Based on your instructions, we logged into the journal's website and submitted the file of the revised manuscript and the file of the point-by-point response to the comments raised by the reviewers in Microsoft Word format.
Appended to this letter is our detailed point-by-point response to the comments raised by the reviewers. We agreed with all the comments. According to the comments raised by the reviewers, we modified manuscript in which the underlined text represents the change made during the revision process.
We take this opportunity to express our gratitude to the reviewers for their constructive and useful remarks. Their comments allowed us to identify areas in our manuscript that needed modification and clarification. We also thank you for allowing us to resubmit a revised copy of the manuscript.
I hope that the revised manuscript is now acceptable for publication in the journal.
Sincerely Yours,
Koji Tanaka, MD PhD
Department of Gastroenterological Surgery
Graduate School of Medicine, Osaka University
2-2 E2 Yamadaoka, Suita
Osaka, 565-0871, Japan.
Tel.: +81-6-6879-3251
Fax: +81-6-6879-3259
E-mail: [email protected]
Point-by-point response to the comments of Reviewer #1
Many thanks for the opportunity to review this manuscript and I commend the authors on taking on this topic. There are a few areas that I think need to be addressed in this manuscript.
Response: We thank the reviewer for the evaluation of our manuscript.
1. Introduction: the introduction is well researched. However, there is minimal reference made to use of vasopressor agents. It is a balancing act between vasopressor use and fluid resuscitation in esophagectomy and this needs to be acknowledged.
Response: We thank the reviewer for the comment. Based on the reviewer’s comment, the association between vasopressor use and fluid resuscitation in esophagectomy is important. This study found that overload fluid balance did not relate to use of vasopressor agents (Table4). Additionally, the vasopressor use did not impact on infusion volume and total fluid balance at POD1 (p=0.860, p=0.189). Thus, we added Results section. (P3 line22-23, P11 line3-5)
Perioperative fluid balance combined vasopressor is an important indicator in the postoperative management of patients after esophageal cancer surgery.
There was no difference in blood transfusion, vasopressor and diuretics utilization rates between the two groups. Also, the vasopressor utilization did not impact on infusion volume and total fluid balance at POD1 (p=0.860, p=0.189).
2. Methods: My main issue here is around how perioperative management was recorded. Although use of epidurals is mentioned, there is no recording of this in the results. Also, there is no mention of intra-operative use of vasopressors (with duration and quantity) or post-operative use of vasopressors. Patients with epidurals have altered sympathetic tone and often end up fluid overloaded: one of the potential benefits of MIE is avoiding this situation.
Response: We thank the reviewer for the comment. The postoperative protocol for epidural analgesia was basically levobupivacaine 0.167% at continuous infusion of 4 mL/h for 5 days after surgery. However, as the reviewer mentioned it, this study did not evaluate a total volume of epidurals including bolus during pain.
Also, the quantity of vasopressor was not assessed because vasopressors used in this study included various agents such as noradrenaline and dopamine and it is difficult to unify the vasopressors quantity. Instead, we found that the vasopressor use did not impact on infusion volume and total fluid balance at POD1 (p=0.860, p=0.189). Based on the reviewer’s comment, Thus, we added limitation in Methods and Results section. (P6 line14-16, P15 line1-2, P11 line3-5)
The postoperative protocol for epidural analgesia was levobupivacaine 0.167% at continuous infusion of 4 mL/h for 5 days after surgery.
Third, this study did not evaluate a total volume of epidurals including bolus during pain in perioperative fluid balance.
There was no difference in blood transfusion, vasopressor and diuretics utilization rates between the two groups. Also, the vasopressor utilization did not impact on infusion volume and total fluid balance at POD1 (p=0.860, p=0.189).
3. Results: Cutting the groups in two based on the median seems a little arbitrary. It would be nice to consider a cutpoint analysis to try determine if there is a threshold where fluid overload is significant. Again, the lack of detail on vasopressors needs to be addressed. How many patients were on pressors intra-op and post op, for how long and what total quantity? How may had epidurals? The language needs clarification that it is the group who are overloaded on POD1 that have complications during their course (not that they have a complication on POD1 as the manuscript seems to suggest).
Response: We thank the reviewer for the comment.
First, as the reviewer mentioned it, this study evaluated the relationship between fluid balance and postoperative complication using the median of perioperative fluid balance. Therefore, we analyzed fluid overload by cut-off point (1000ml, 2000ml, 3000ml, 4000ml and 5000ml, respectively), and 3000ml overload balance had most high rate of postoperative acute pneumonia and anastomotic leakage in cutoff analysis (33% and 15%). Moreover, the present study validated that the influence of fluid balance at POD1 on postoperative acute pneumonia and anastomotic leakage by receiver operating characteristic (ROC) analysis. The cut off value using ROC curve for the fluid balance at POD1 in postoperative acute pneumonia and anastomotic leakage was 3080mL and 3094ml which is similar to median value, respectively (acute pneumonia; sensitivity of 0.71 and specificity of 0.44, anastomotic leakage; sensitivity of 0.88 and specificity of 0.46) (Supplemental Figure1). Therefore, we consider that the median value was reasonable threshold by cut-point analysis and ROC curve.
Second, as we mentioned earlier, intra-operative use of vasopressors with quantity were not assessed because vasopressors used in this study included various agents such as noradrenaline and dopamine and it is difficult to standardize the vasopressors with duration and quantity. However, this study found that the use of vasopressor did not effect on infusion volume and total fluid balance at POD1 (p=0.860, p=0.189).
Third, the present study did not assess epidural volume. In this study, the postoperative protocol for epidural analgesia was levobupivacaine 0.167% at continuous infusion of 4 mL/h for 5 days after MIE. However, we did not investigate a total volume of epidurals including bolus during pain.
Forth, thank you for your indication. We modified the manuscript which clarified that the fluid balance affected on complications during course from postoperative to discharge.
Thus, we added limitation in Materials and Methods, Results, Discussion and Supplemental Figure 1 section. (P9 line12-16, P11 line3-5, P6 line14-16, P7 line5-6, P9 line1-3)
Also, the cutoff value using ROC curve for the fluid balance at POD 1 in postoperative acute pneumonia and anastomotic leakage was 3080mL and 3094ml which was similar to median value, respectively (acute pneumonia; sensitivity of 0.71 and specificity of 0.44, anastomotic leakage; sensitivity of 0.88 and specificity of 0.46) (Supplemental Figure1).
There was no difference in blood transfusion, vasopressor and diuretics utilization rates between the two groups. Also, the vasopressor utilization did not impact on infusion volume and total fluid balance at POD1 (p=0.860, p=0.189).
The postoperative protocol for epidural analgesia was levobupivacaine 0.167% at continuous infusion of 4 mL/h for 5 days after surgery.
The complications during course from postoperative to discharge were defined as. Clavien-Dindo classification grade ≥ II (18).
There was no difference in overall complications during course from postoperative to discharge (Clavien–Dindo classification grade ≥ II) between the two groups during surgery and at POD 1 and POD 2, respectively.
4. The discussion is fine, but will need to be rewritten taking into account the points above. There are several important points which need to be taken into account.
Response: We thank the reviewer for the comment and modified the above section.

Reviewer 2 Report
The authors present an important contribution to the management of patients during and after esophagectomy. I have the following comments:
- Is it possible to identify indications for application of fluids or blood products? (e. g. hypotension, low urine output)
- The authors provided the use of vasopressors. However, the dosage of vasopressor therapy would be more interesting, and could be stratified in low and high (cutoff for example at 0.15 µg/kg noradrenaline)
- The authors reported on urine output, however, failed to present the rate of postoperative acute kidney injury and the need for dialysis. In my opinion, these parameters are important to adequately estimate the need for application of fluids. In that regard, the reader should be informed about preexisting renal comorbidities.
- In the limitations, the authors rightly indicate that fluid balance during surgery may differ depending on the anesthesiologist. I think this also applies for the attending doctor on the ICU.
Author Response
Thank you for your e-mail of May 18, 2022. We are pleased to have a chance to resubmit our revised manuscript in Journal of Clinical Medicine.
Based on your instructions, we logged into the journal's website and submitted the file of the revised manuscript and the file of the point-by-point response to the comments raised by the reviewers in Microsoft Word format.
Appended to this letter is our detailed point-by-point response to the comments raised by the reviewers. We agreed with all the comments. According to the comments raised by the reviewers, we modified manuscript in which the underlined text represents the change made during the revision process.
We take this opportunity to express our gratitude to the reviewers for their constructive and useful remarks. Their comments allowed us to identify areas in our manuscript that needed modification and clarification. We also thank you for allowing us to resubmit a revised copy of the manuscript.
I hope that the revised manuscript is now acceptable for publication in the journal.
Sincerely Yours,
Koji Tanaka, MD PhD
Department of Gastroenterological Surgery
Graduate School of Medicine, Osaka University
2-2 E2 Yamadaoka, Suita
Osaka, 565-0871, Japan.
Tel.: +81-6-6879-3251
Fax: +81-6-6879-3259
E-mail: [email protected]
Point-by-point response to the comments of Reviewer #2
The authors present an important contribution to the management of patients during and after esophagectomy. I have the following comments:
Response: We thank the reviewer for the positive evaluation of our manuscript.
- Is it possible to identify indications for application of fluids or blood products? (e. g. hypotension, low urine output)
Response: We thank the reviewer for the comment. This study found that the fluid overload significantly related to less urine output. Therefore, we suggest that fluid balance under low urine output needs to be strictly controlled after MIE and believe that our findings will be useful in clinical therapeutic decisions for esophageal cancer. Thus, we modified Results section. (P11 line1-3)
On the other hand, the higher group had significantly less urine output than the lower group, although there was no difference in drainage output between the two groups.
- The authors provided the use of vasopressors. However, the dosage of vasopressor therapy would be more interesting, and could be stratified in low and high (cutoff for example at 0.15 µg/kg noradrenaline)
Response: We thank the reviewer for the comment. As the reviewer. the present study investigated that use of vasopressors in two groups at POD1. However, intra-operative use of vasopressors with quantity were not assessed because vasopressors used in our study included various agents such as noradrenaline and dopamine and it is difficult to unify the vasopressors with duration and quantity. Instead, this study found that the use of vasopressor did not have influence on infusion volume and total fluid balance at POD1 (p=0.860, p=0.189). Thus, we modified Results section. (P11 line3-5)
There was no difference in blood transfusion, vasopressor and diuretics utilization rates between the two groups. Also, the vasopressor utilization did not impact on infusion volume and total fluid balance at POD1 (p=0.860, p=0.189).
- The authors reported on urine output, however, failed to present the rate of postoperative acute kidney injury and the need for dialysis. In my opinion, these parameters are important to adequately estimate the need for application of fluids. In that regard, the reader should be informed about preexisting renal comorbidities.
Response: Thank for the comment. We investigate postoperative acute kidney injury and found that acute kidney injury had no difference between the two group. Moreover, there was no patient with acute kidney failure underwent dialysis after surgery. Thus, we added limitation in Methods, Results and Reference section and Table 1. (P7 line22-24, P9 line9-11, P19 line17-18)
Also, acute kidney injury (AKI) was defined by increased serum creatinine (Cr) and urine output using Kidney Disease Improving Global Outcomes (KDIGO) clinical practice guidelines (26).
There was no difference in arrythmia, recurrent nerve palsy, chylothorax, pulmonary embolism or AKI (KDIGO clinical practice guidelines Stage ≥ 1) between the two groups and no patients with acute kidney failure underwent dialysis after surgery.
- Khwaja A. KDIGO clinical practice guidelines for acute kidney injury. Nephron Clin Pract. 2012, 120, 179-184.
- In the limitations, the authors rightly indicate that fluid balance during surgery may differ depending on the anesthesiologist. I think this also applies for the attending doctor on the ICU.
Response: We thank the reviewer for the comment. As the reviewer mentioned it, the perioperative fluid balance may differ depending on the attending doctor on the ICU as well as anesthesiologist. Thus, we modified limitation in Discussion section. (P14 line22-24)
Second, anesthesiologists and ICU doctors who participated in this study were random, not constant. Hence, indications for fluid management or blood products differed by anesthesiologists and ICU doctors.
